# Dual-comb photoacoustic spectroscopy

Jacob T. Friedlein[1], Esther Baumann[1,2], Kimberly A. Briggman[1], Gabriel M. Colacion[1,2,3], Fabrizio R. Giorgetta[1,2], Aaron M. Goldfain [1], Daniel I. Herman[1,2], Eli V. Hoenig[1,2,4], Jeeseong Hwang[1], Nathan R. Newbury[1], Edgar F. Perez[1,2,5], Christopher S. Yung [1], Ian Coddington[1] & Kevin C. Cossel [1✉]

Spectrally resolved photoacoustic imaging is promising for label-free imaging in optically scattering materials. However, this technique often requires acquisition of a separate image at each wavelength of interest. This reduces imaging speeds and causes errors if the sample changes in time between images acquired at different wavelengths. We demonstrate a solution to this problem by using dual-comb spectroscopy for photoacoustic measurements. This approach enables a photoacoustic measurement at thousands of wavelengths simultaneously. In this technique, two optical-frequency combs are interfered on a sample and the resulting pressure wave is measured with an ultrasound transducer. This acoustic signal is processed in the frequency-domain to obtain an optical absorption spectrum. For a proof-of-concept demonstration, we measure photoacoustic signals from polymer films. The absorption spectra obtained from these measurements agree with those measured using a spectrophotometer. Improving the signal-to-noise ratio of the dual-comb photoacoustic spectrometer could enable high-speed spectrally resolved photoacoustic imaging.

[1] National Institute of Standards and Technology, Applied Physics Division, 325 Broadway, Boulder, CO 80305, USA. [2] Department of Physics, University of Colorado, Boulder, CO 80309, USA. [3] Present address: Optical Science and Engineering, University of New Mexico, 1313 Goddard, SE, Albuquerque, NM 87106, USA. [4] Present address: Pritzker School of Molecular Engineering, University of Chicago, 5640 South Ellis Avenue, ERC 387, Chicago, IL 60637, USA. [5] Present address: Institute for Research in Electronics and Applied Physics, University of Maryland, 8279 Paint Branch Drive, College Park, MD 20742-3511, USA. ✉email: kevin.cossel@nist.gov

In vivo and in vitro medical imaging and spectroscopy face the fundamental challenge of strong optical scattering in biological tissues. This challenge has led researchers to use the photoacoustic (PA) effect, where localized light absorption generates acoustic waves that can be detected from deep within tissue[1–4]. PA measurements have an advantage over pure optical methods because light only has to travel through the sample to the absorber to create an acoustic wave, but the light does not need to travel back out of the sample to a detector. Other advantages of PA measurements are that scattered optical illumination can still induce a PA signal, and the generated acoustic waves are only weakly scattered by tissue[5]. In addition, by measuring the PA response versus excitation wavelength, it is possible to identify and quantify substances based on their unique absorption features. For example, researchers have used this method, called PA spectroscopy, to quantify glucose levels in vivo[6–8]. A more advanced technique for medical diagnostics and functional imaging, called multispectral optoacoustic tomography (MSOT) or photoacoustic tomography (PAT), has also been demonstrated. This technique provides images of multiple exogenous and endogenous species at multiple wavelengths and uses spectral unmixing algorithms to determine the spatial distribution of each species[2,9,10]. This technique has been used to differentiate adipose tissue from cholesterol[11,12], measure intramuscular fat distributions[13], map tumor size and shape[14,15], identify white matter loss in spinal cord injuries[16], quantify blood oxygen saturation[17,18], and even to construct a label-free molecular map of a mouse fibroblast cell[19].

One challenge of PA spectroscopy, or PAS, measurements is that they require a broadband optical excitation source with a spectrally resolved detection or illumination method. Traditionally, PA researchers have used optical parametric oscillators or other tunable lasers, which require sequential image acquisition over narrow optical-frequency bands. Although it is possible in simple ratiometric PAS to use only a couple of optical-frequency bands, measuring in more bands allows the identification of additional species and improves detection sensitivity[20]. However, sequentially scanning the optical frequency can be time consuming and can lead to potential issues with sample drift or damage between images at different optical frequencies[9,21]. Broadband thermal sources using a Fourier transform spectrometer (FTS)[22–25] for optical-frequency resolution have been used to overcome sequential scanning, but the low optical power and slow scanning delay arm have limited the application of these systems. Alternatively, broadband supercontinuum coherent laser sources may provide higher power than a thermal source, but still require either sequential frequency scanning or physical scanning of an FTS delay arm and can suffer from high laser intensity noise. Optical frequency combs[26–28] are an alternative light source for broadband PAS and multispectral PAT because they simultaneously generate thousands of discrete optical-frequency bands. These frequency bands are called comb teeth because, like the teeth of a comb, they are evenly spaced and very narrow, with absolute linewidths below 120 kHz[29]. The comb teeth of a frequency comb are temporally and spatially coherent and propagate in a single spatial mode with high brightness. Recent work has shown that frequency combs are a promising light source for PAS. Researchers have demonstrated highly sensitive and precise measurements of methane absorption spectra with frequency-comb-based PAS[30,31]. These demonstrations relied on a FTS for spectral selectivity and thus required physical scanning of the optical pathlength. It is possible, however, to eliminate the need for the FTS altogether by using a second comb like a local oscillator. By exploiting the coherence of optical-frequency combs, one can capture an entire absorption spectrum on a single photodetector without wavelength or delay arm scanning, as is done in dual-comb spectroscopy (DCS)[32]. DCS provides broad spectral coverage, high sensitivity, and high spectral resolution for identification of multiple chemical species[33–37]. It also can provide rapid spectral measurements for monitoring dynamic biological processes[38], as well as for coherent Raman imaging[39,40]. These advantages should be applicable to PAS using a technique analogous to DCS. This method of dual-comb PAS (DCPAS), which was suggested in ref. [31], can be used across broad spectral windows without sequential wavelength scanning and can be implemented with a compact, stable, and robust dual-comb spectrometer.

In this article, we report an experimental demonstration of DCPAS. We use two self-referenced fiber-laser frequency combs to measure the PA spectra of thick polymer films over an ≈15 THz spectral bandwidth centered at 173 THz. With coherent averaging we obtain spectrally resolved PA responses of polydimethylsiloxane (PDMS) and paraffin samples immersed in water. These experiments reveal that dual-comb optical excitation can generate PA signals and that these signals can be used to detect and identify organic materials.

## Results

**DCPAS concept.** Figure 1 describes the concept of DCPAS. The DCPAS system is based on laser frequency combs whose frequency-domain output is tens of thousands of evenly spaced narrow lines or comb "teeth". These teeth are spaced at the frequency-comb's pulse repetition rate, $f_{rep}$, ((i) in Fig. 1)[27]. In the dual-comb approach, we use two such frequency combs where the repetition rate of comb 2, $f_{rep,2}$, is slightly larger than the repetition rate of comb 1, $f_{rep,1}$, such that $f_{rep,2} - f_{rep,1} = \Delta f_{rep}$. As a result, the frequencies of subsequent pairs of comb teeth are offset by increasing increments of $\Delta f_{rep}$, as illustrated in Fig. 1. The optical intensity of the combined combs is given by $I_0 = |E_1 + E_2|^2$, where $E_1$ is the electric field of the light from comb 1 and $E_2$ is the electric field of the light from comb 2. This intensity contains a multi-heterodyne term corresponding to the beat notes of each, increasingly offset, pair of comb teeth. Assuming the zeroth tooth of comb 1, at frequency $\nu_{1,0}$, overlaps with the zeroth tooth of comb 2, at frequency $\nu_{2,0} = \nu_{1,0}$, then the beat notes occur at radio frequencies, $f_i$, given by

$$
\begin{aligned}
f_1 &= \nu_{2,1} - \nu_{1,1} &&= [\nu_{2,0} + 1 \times f_{rep,2}] - [\nu_{1,0} + 1 \times f_{rep,1}] &&= 1 \times \Delta f_{rep} \\
f_2 &= \nu_{2,2} - \nu_{1,2} &&= [\nu_{2,0} + 2 \times f_{rep,2}] - [\nu_{1,0} + 2 \times f_{rep,1}] &&= 2 \times \Delta f_{rep} \\
f_3 &= \nu_{2,3} - \nu_{1,3} &&= [\nu_{2,0} + 3 \times f_{rep,2}] - [\nu_{1,0} + 3 \times f_{rep,1}] &&= 3 \times \Delta f_{rep} \quad (1) \\
\vdots &= \quad\vdots &&= \qquad\qquad\vdots &&= \quad\vdots \\
f_n &= \nu_{2,n} - \nu_{1,n} &&= [\nu_{2,0} + n \times f_{rep,2}] - [\nu_{1,0} + n \times f_{rep,1}] &&= n \times \Delta f_{rep}
\end{aligned}
$$

As the optical frequencies $\nu_{i,j}$ are typically 10 to 12 orders of magnitude larger than $\Delta f_{rep}$, $\nu_{1,i}$ is approximately the same as $\nu_{2,i}$, and it is quite accurate to say that light at the optical-frequency $\nu_i = \nu_{1,i} \approx \nu_{2,i}$ is amplitude modulated at the radio frequency $f_i$. When the sample absorbs light from a pair of comb teeth at optical-frequency $\nu_i$, it produces a PA pressure wave with an amplitude, $p$, proportional to the amount of light absorbed by the sample ((iii) in Fig. 1)[41–43]. This proportionality is given by $p \propto I_0(\nu_i) \times [1 - \exp(-\mu_A(\nu_i) \times L)] \times \beta$, where $I_0(\nu_i)$ is the intensity of the dual-comb excitation at optical-frequency $\nu_i$, $\mu_A(\nu_i)$ is the sample's base-e absorption coefficient at optical-frequency $\nu_i$, $L$ is the sample thickness that contributes to the PA signal, and $\beta$ is a proportionality constant that depends on sample material and geometry. The amplitude of this pressure wave is modulated at

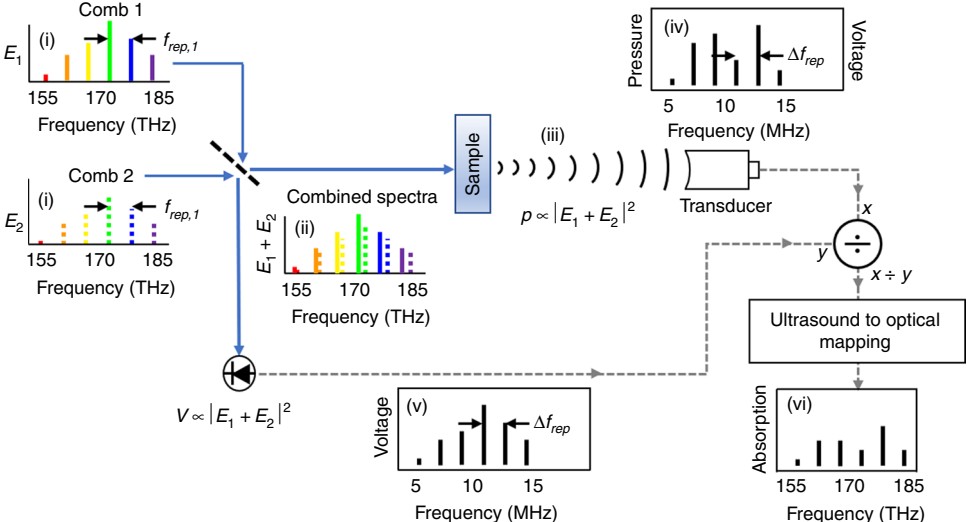

**Fig. 1 Schematic of dual-comb photoacoustic spectroscopy.** Two frequency combs (i) are combined on a beam splitter and focused onto a sample. The sample absorbs this combined light (ii) and generates localized thermoelastic energy, which results in a PA pressure wave upon transmitting into the water (iii). An ultrasound transducer detects and converts this pressure wave to a voltage signal whose Fourier transform yields an ultrasound signal (iv). Simultaneously, the combined comb light from the other beam-splitter branch is focused onto a photodetector to record the optical excitation spectrum as in DCS (v). The spectrally resolved DCPAS response is normalized to the intensity of the DCS excitation spectrum (v). Then a one-to-one mapping between the ultrasound and optical frequencies is applied to yield the sample's optical absorption spectrum (vi). Note that the comb spectra are not to scale as there are 95,000 comb teeth within the ≈15 THz spectral bandwidth used in these experiments.

the multi-heterodyne frequency $f_i$, which can be several orders of magnitude less than the comb repetition frequencies, $f_{\rm rep}$. When the sample absorbs light from many pairs of comb teeth, at optical frequencies $\nu_0, \nu_1. \nu_2, \ldots, \nu_n$, it produces a PA pressure wave with frequency components centered at the corresponding modulation frequencies $f_0, f_1, f_2, \ldots f_n$. As there is a one-to-one correlation between the frequency $\nu_i$ of the absorbed light and the resulting acoustic modulation frequency, $f_i$, one can infer the sample's optical absorption spectrum by analyzing the frequency-domain PA signal. As every comb tooth is simultaneously incident on the sample and every PA frequency is simultaneously recorded, this technique allows parallel acquisition of thousands of spectral elements and multiplexed measurement of the sample's absorption spectrum.

It is also useful to consider the time-domain picture. In the time-domain, the DCPAS signal is a series of interferograms, just as in DCS. Each interferogram reflects the time-dependent optical intensity caused by the interference between successive pulses from the two offset frequency combs. In DCPAS, this time-dependent optical intensity leads to a time-dependent amplitude modulation of the PA pressure wave, which we detect via an ultrasound transducer. The pressure waves are detected by the transducer to generate a corresponding time-domain voltage interferogram. We then convert these interferograms to frequency-domain spectra by a Fourier transform to generate the multi-heterodyne acoustic signal discussed above ((iv) in Fig. 1). This acoustic signal is normalized by the optical excitation spectrum recorded by a photodetector ((v) in Fig. 1). Then the acoustic frequencies are mapped to optical frequencies using the known parameters of the combs[32]. This yields the sample's optical absorption spectrum ((vi) in Fig. 1).

One advantage of DCPAS is that the bandwidth and center frequency of the generated acoustic signal, corresponding to the amplitude modulation on the optical signal, can be adjusted independently of the optical bandwidth and center frequency. This adjustment is made in a controlled fashion by tuning the frequency locking of the two combs to adjust $\nu_{2,0}$, $\nu_{1,0}$, and $\Delta f_{\rm rep}$, and therefore, the multi-heterodyne frequencies $f_i$. This allows

any optical spectrum to generate a PA signal that can be detected by an ultrasound transducer despite the transducer's limited bandwidth. In this demonstration, we use lasers with ≈160 MHz repetition rates and set the difference in repetition rates (tooth spacings), $\Delta f_{\rm rep} = f_{\rm rep,2} - f_{\rm rep,1}$, to be 66.81 Hz. This allows us to map the roughly 15 THz optical bandwidth (full width at −10 dB) into an acoustic bandwidth of (15 THz) × ($\Delta f_{\rm rep}/f_{\rm rep}$) ≈6.3 MHz[32] while centering this modulation signal at the peak of the transducer responsivity, 7.5 MHz. In order to maintain this mapping, as well as the optical coherence required for long-term averaging of the signals, both combs are fully stabilized in the manner described in ref. [44].

**Experimental setup.** Figure 2 shows the experimental setup for DCPAS. We use two self-referenced erbium fiber frequency combs at 160 MHz repetition rate. The output of each comb is amplified and filtered to generate light in a 15-THz-wide band centered at ≈173 THz, selected to coincide with low water absorption and the first overtone absorption of C–H bonds. This wavelength region has previously been used to image arterial plaques (fatty tissue) in the presence of blood, to distinguish protein and fat, and to image white matter in a spinal cord[4,13]. A total of about 24 mW of comb light is incident on the samples. The PA signals generated by the samples then propagate through a water bath to a polytetrafluoride ultrasound transducer with a frequency response that is peaked at 7.5 MHz with an ≈7.5 MHz bandwidth (see Supplementary Fig. 3). The PA and photodetected signals are coherently summed in a data acquisition system previously used for mid-infrared DCS[34,45]. The methods section provides more detail of the experimental setup.

**DCPAS experimental results.** Figure 3 shows the DCPAS measurement results from a sample consisting of vertically aligned carbon nanotubes (VACNTs) embedded in PDMS (polydimethylsiloxane). This sample was chosen because VACNTs have strong optical absorption at ≈170 THz[46] and

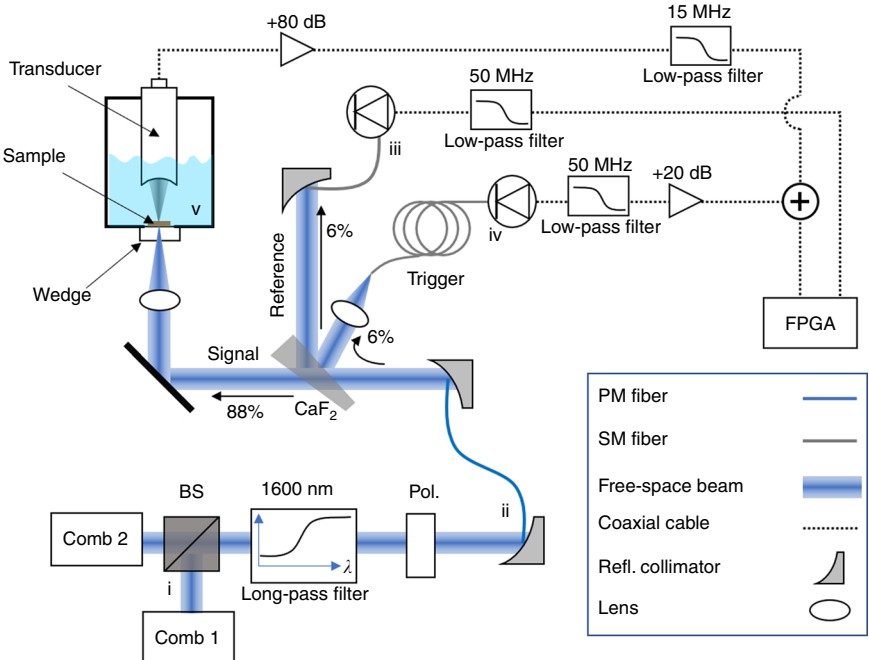

**Fig. 2 Experimental setup for dual-comb photoacoustic spectroscopy.** Two frequency combs (lower left) are combined on a beam-splitter and filtered. The combined beam is focused onto a sample in a water bath (upper left). A focused ultrasound transducer measures the photoacoustic pressure waves emitted by the sample when it absorbs light. The transducer output is amplified and filtered before being digitized with a field-programmable gate array (FPGA). Approximately 6% of the combined beam is picked off by a calcium-fluoride wedge (center) and used as a reference signal. Another 6% of the beam is used for triggering the data acquisition system (right-center). Additional details are discussed in the methods section.

therefore generate a strong PA response when illuminated. This provides a clear demonstration that dual-comb optical excitation (Fig. 3a) induces a PA response (Fig. 3b). Furthermore, the shape of the PA response reveals that the PA pressure wave is an interferogram that corresponds to the optical interferogram incident on the sample. The correlation between the optical excitation and the PA response can be probed further by examining the frequency-domain representation of these signals (Fig. 3c). To obtain the frequency-domain signals, the time-domain signals are apodized with a 4.8 µs window, zero-padded outside of the apodization window, and Fourier transformed. The Fourier transform yields the spectrally resolved optical excitation and PA response at acoustic frequencies corresponding to the beat frequencies between the teeth of the two frequency combs (top axis of Fig. 3c). To convert these acoustic or ultrasound frequencies, $f_{US}$, to the optical frequencies, $\nu_{optical}$, of the comb teeth that generated the signal, we linearly scale the frequency according to:

$$\nu_{optical} = \nu_0 + \frac{f_{rep,1} + f_{rep,2}}{2 \times (f_{rep,2} - f_{rep,1})} \times f_{US}, \qquad (2)$$

where $f_{rep,1}$ is the repetition rate of comb 1, $f_{rep,2}$ is the repetition rate of comb 2, and $\nu_0$ is a constant offset calculated from the comb repetition rates and carrier-envelope offset frequencies. With the 4.8 µs apodization window used here, the spectral resolution of the measurement is ≈500 GHz or ≈5 nm (due to the zero-padding, the spectrum is effectively smoothed to 500 GHz).

The spectrally resolved PA response shows how a sample's absorption coefficient depends on optical frequency. At frequencies where the sample absorbs strongly, there is a strong PA response; whereas, at frequencies where the sample does not absorb, there is no PA response. Therefore, if a sample absorbs uniformly at all frequencies contained in the optical

excitation, the spectrally resolved PA response should have the same shape as the optical excitation spectrum. The spectrally resolved PA response of the VACNT sample (Fig. 3c, red line) has a frequency dependence that is similar to the optical excitation spectrum (Fig. 3c, black line). This similarity shows that the PA response is dependent on the optical excitation spectrum. The relatively weaker amplitude of the PA response for optical frequencies $\nu_{optical} \lesssim 166\,THz$ and $\nu_{optical} \gtrsim 176\,THz$ is due to a roll-off in the transducer's responsivity away from its peak responsivity at $f_{US} = 7.5\,MHz$ (which corresponds to $\nu_{optical} = 171.2\,THz$). There are small (around 20%) variations in the PA response relative to the optical excitation spectrum for optical frequencies between 166 THz and 176 THz whose origins are unknown, but could be caused by sample inhomogeneities. Despite these variations, the VACNT target was useful for characterizing the DCPAS system and for demonstrating that dual-comb excitation can induce a PA interferogram. However, the VACNT's absorption spectrum is not representative of a biological material because it lacks a characteristic C-H overtone in the 175-THz band.

To observe the C-H overtone absorption, we measured two different polymer thin films: polydimethylsiloxane (PDMS) and paraffin, both of which are much more weakly absorbing in this spectral region than the VACNT sample. The PA response of PDMS and paraffin samples are shown in Fig. 4a, b, respectively, for several different averaging times. As the polymers absorb more weakly than the VACNTs and are less efficient at generating acoustic waves, they produce a DCPAS signal that is 50× and 100× weaker than the VACNTs, as can be seen from comparing Fig. 3c to Fig. 4a, b, respectively. The spectral features of the PDMS response are somewhat distinguishable over the noise with as little as 4 min of signal averaging and are easily distinguishable with 20 min of averaging. On the other hand, the spectral features of the paraffin response are readily apparent even at 4 min of

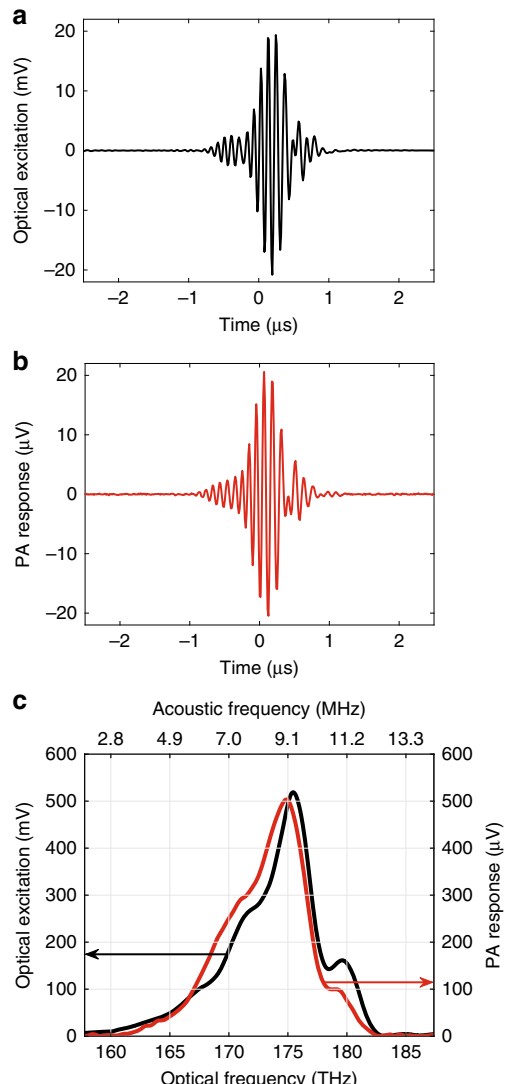

**Fig. 3 Dual-comb photoacoustic spectroscopy of a vertically aligned carbon nanotube film embedded in polydimethylsiloxane. a** Time-domain interferogram for the optical excitation measured by the ac-coupled photoreceiver. **b** Time-domain interferogram for the PA response at the transducer output (obtained by dividing the measured signal by the amplifier gain). These time-domain signals are obtained with 60 seconds of real-time coherent averaging. **c** Optical excitation spectrum (black line, left axis) and spectrally resolved PA response (red line, right axis). The spectrally resolved data are obtained by first apodizing the time-domain signal with a 4.8-μs window (zero-padded outside of window), then Fourier transformation, and finally scaling to optical frequencies (see equation (2)). The upper axis in **c** shows the beat frequencies or acoustic frequencies before scaling to optical frequencies. In **b** an acoustic propagation time, $t_{acoustic} = 14\,\mu s$, was subtracted to account for the propagation time of the acoustic waves in water between the sample and transducer. The average optical power incident on the sample is $\approx 24\,mW$.

averaging due to the approximately two times stronger signal from the paraffin film.

Figure 4c, d compare these PA responses to the samples' optical absorption spectra measured with a spectrophotometer. In order to make this comparison, we need to account for several effects inherent to the DCPAS measurement. As mentioned in the introduction, the amplitude of the PA pressure wave is proportional to the sample's optical

absorbance, the intensity of the optical excitation at the sample, and other things such as acoustic impedance mismatch and the sample's acoustic response. The measured PA response is proportional to the pressure wave's amplitude multiplied by the transfer function of the transducer[47,48]. For weakly absorbing samples like the paraffin and PDMS in these experiments, the exponential in Beer's law can be linearized, and we can write the measured PA response as

$$V_{PA}\left(\nu_{optical}\right) \approx \beta \times \mu_A\left(\nu_{optical}\right) \times I_0\left(\nu_{optical}\right) \times H_{transducer}\left(\nu_{optical}\right),$$
(3)

where $V_{PA}$ is the measured PA response as a function of optical frequency, $\nu_{optical}$, $\beta$ is a proportionality factor to convert from sample absorption to PA pressure, which may be material and geometry dependent, $\mu_A(\nu_{optical})$ is the sample's optical absorption coefficient at optical frequency $\nu_{optical}$, $I_0(\nu_{optical})$ is the intensity of the optical excitation at optical frequency $\nu_{optical}$, and $H_{transducer}(\nu_{optical})$ is the ultrasound transducer's frequency-dependent responsivity after scaling to optical frequencies according to Eq. (2). An apparent optical-frequency dependence of $\beta$ could arise from acoustic-frequency-dependent pressure generation or attenuation; however, here we assume that $\beta$ is constant over the acoustic frequencies used for the amplitude modulation in these experiments. $H_{transducer}$ is obtained from calibration curves provided by the manufacturer (Supplementary Fig. 3), and $I_0(\nu_{optical})$ is measured via the photodetected DCS spectrum. Thus, the normalized PA response of a sample is

$$\widehat{V_{PA}}\left(\nu_{optical}\right) = \frac{V_{PA}\left(\nu_{optical}\right)}{I_0\left(\nu_{optical}\right) \times H_{transducer}\left(\nu_{optical}\right)} = \beta \times \mu_A\left(\nu_{optical}\right).$$
(4)

Figure 4c, d show the result of normalizing the polymers' PA responses according to Eq. (4). For comparison, these figures also show the optical absorbance spectra, $\propto \mu_A(\nu_{optical})$, of PDMS and paraffin measured in a transmission-mode spectrophotometer. The normalized PA responses reflect the absorbance spectra from the spectrophotometer measurements. The DCPAS measurement reveals prominent PDMS absorbance peaks at 171.6 THz, 176.0 THz, 177.4 THz. The location of these peaks are in good agreement with the spectrophotometer absorbance measurement and previously published results[49,50]. The normalized PA response of paraffin contains well-defined absorbance peaks at 169.8 THz and 173.1 THz. Although we could not find literature showing paraffin absorbance spectra in the 165 THz to 180 THz window, the absorbance peaks from the DCPAS measurement closely match the peaks from our spectrophotometer measurement, and the peak at 173.1 THz agrees well with the location of a reported absorbance peak[51].

The results shown in Fig. 4 prove that DCPAS is sensitive to the absorption spectra of polymer samples. However, the signal-to-noise ratio (SNR) of this proof-of-concept DCPAS system is low for weakly absorbing features such as the C-H overtone bands in these polymer films. Thus, long averaging times are currently required to obtain spectrally resolved responses like those in Fig. 4. Figure 5a shows the time-domain SNR versus averaging time for both VACNTs and the polymer samples. (The time-domain SNR is defined in the Methods section and Supplementary Note 1.) In all cases the SNR is proportional to the square root of averaging time, as expected for white noise. The VACNT sample requires 4 orders of magnitude less averaging time than the polymer samples to reach the same SNR because the VACNTs produce an ≈100× stronger PA signal than the polymers. Figure 5b shows that the

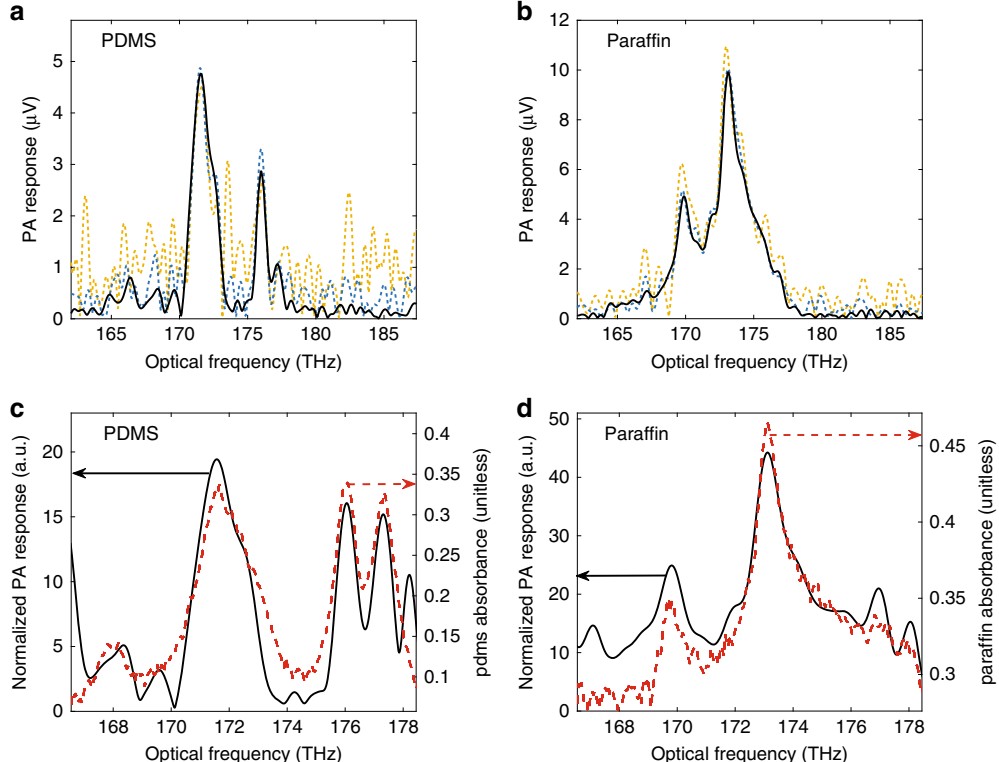

**Fig. 4 Spectrally resolved dual-comb photoacoustic responses of polymer films. a**, **b** show the PA response for PDMS (**a**) and paraffin (**b**) obtained by Fourier transformation of the time-domain signal at the transducer output (obtained by dividing the measured signal by the amplifier gain) followed by scaling to optical frequencies (Eq. (2)). The averaging times are 4 min, 20 min, and 120 min for the yellow-dotted, blue-dotted, and black-solid traces, respectively. **c**, **d** show the normalized PA response from Eq. (4) (solid black) compared to the optical absorbance (base-e absorbance units) measured with a spectrophotometer (dashed red) for PDMS (**c**) and paraffin (**d**). See "Methods" section and Supplementary Note 3 for details about sample preparation. The optical excitation power is ≈24 mW for both the PDMS and paraffin measurements.

SNR is linearly proportional to the power of the optical excitation. This proportionality indicates that the SNR of the system is limited by detector noise and not subject to saturation effects even for the strongly absorbing VACNT sample. This lack of saturation suggests that higher SNR measurements of organic films are feasible with a higher-power dual-comb source.

### Discussion
The reported results highlight the potential of DCPAS for label-free imaging without spectral scanning. As shown in Fig. 5, a high SNR ( >300) can be reached through coherent averaging. However, the averaging times required for high spectral SNR are likely too long to be useful, especially for medical imaging applications, and it is useful to consider potential routes to reduce the acquisition time.

First, the signal strength could be improved by operating at a wavelength range with larger molecular absorption cross-sections. For example, the PA signal from the VACNT sample has a much higher SNR than that of the PDMS sample because the VACNTs absorb light much more strongly than the PDMS in this wavelength region. In this proof-of-principle experiment, we focused on the optical window at ≈173 THz because it was easily accessible with our laser system, contains first overtone absorption by C–H bonds, and has lower water absorption than adjacent frequency bands[13]. However, mid-infrared frequency combs[34,35,38,52] could measure much stronger fundamental C-H rovibrational transitions. These stronger absorption features (up to several orders of magnitude) could provide a proportional increase in the PA signal

strength and a corresponding quadratic decrease in the required averaging time. However, strong water absorption would limit mid-infrared DCPAS to <100 µm penetration depths in biological tissue because the 1/e penetration depth is only 32 µm for excitation at $\nu_{\text{optical}} = 60$ THz[53]. Nevertheless, mid-infrared DCPAS could be useful for measuring thin samples in the same way that pulsed PAS can measure thin samples at mid-infrared wavelengths[8]. Moving to visible/near-infrared wavelengths would enable measurements of oxygenated and deoxygenated hemoglobin. This visible/near-infrared hemoglobin absorption is about ten times stronger than lipid absorption at around 173 THz without any competing absorption from water[4]. Additionally, other exogenous agents and dyes with strong absorption features could be measured with visible/near-infrared wavelengths[9].

The acquisition time could also be reduced by use of a higher comb repetition frequency. In DCPAS, the fundamental sample point spacing of the measured spectrum is set by the comb repetition rate, $f_{\text{rep}}$. In this demonstration, the spectral sample spacing is $f_{\text{rep}} = 160$ MHz, much finer than any absorption feature of interest. For this reason, we apply a 4.8-µs apodization window before the Fourier transformation. This apodization improves the spectral SNR by a factor of $\sqrt{500\,\text{GHz}/160\,\text{MHz}} \approx 56$ and smooths the frequency-domain response to a spectral resolution of ≈500 GHz. This 500-GHz resolution is acceptable for the envisioned applications because the absorption features in tissue samples are typically 100 s of GHz to 1000 s of GHz wide. For instance, the full-widths at half-maximum of lipid absorption spectra at 172 THz are ≈6000 GHz, and the absorption peaks of plaque lipids and adipose lipids are separated by about 1200

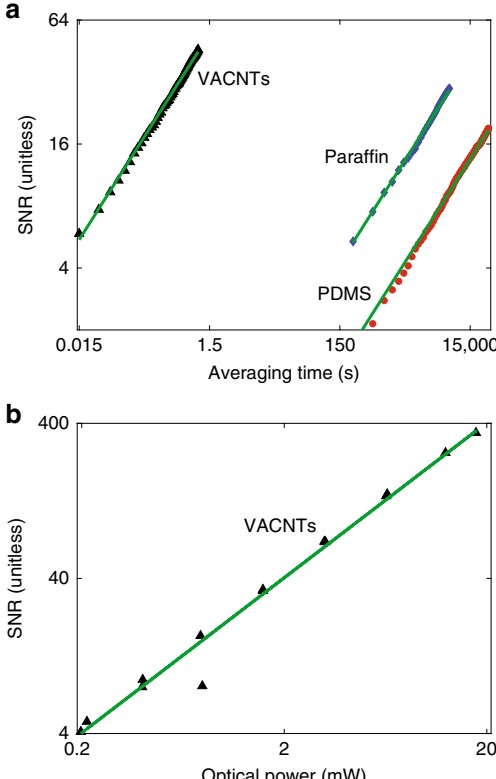

**Fig. 5 Signal-to-noise ratio (SNR) of dual-comb photoacoustic spectroscopy. a** Time-domain SNR of the PA signal as a function of averaging time for the VACNTs at an incident optical power of 18 mW (black triangles), paraffin at an incident power of 24 mW (blue diamonds), and PDMS at an incident power of 24 mW (red circles). The solid green lines are fits to the data according to SNR $= A \times \sqrt{\tau}$, where $\tau$ is the averaging time and $A$ is a fit parameter. **b** Time-domain SNR of the PA signal for the VACNT target as a function of optical excitation power at a fixed averaging time of 61 s (triangles) and a linear fit to optical excitation power (green line). In these measurements, spectral SNRs are roughly the same as time-domain SNRs. For comparison, a time-domain SNR of 32 for paraffin transforms to a peak spectral SNR of 45. This scaling changes slightly depending on the spectrum. As discussed in Supplementary Note 1, the spectral SNR at optical frequency $\nu$ is defined as the mean magnitude of the PA response at $\nu$ divided by the standard deviation of the PA response at $\nu$ over many repeated measurements.

GHz[12]. However, rather than smoothing the frequency-domain response via apodization, it would be better if the fundamental sample spacing, $f_{rep}$, matched the desired resolution. In other words, there is an SNR penalty because the fundamental spectral sample spacing is not matched to the desired resolution. This SNR penalty can be understood by recognizing that a wider tooth-spacing, with a fixed total power, results in a greater power per comb tooth. Taking this into account, one finds that the required averaging time should scale linearly with $1/f_{rep}$ for a fixed output power and fixed SNR up until the repetition rate matches the desired resolution[54]. It is possible to generate frequency combs with 10 GHz repetition rates from a mode locked laser[55,56], quantum cascade laser-based combs[38], or the use of electro-optic frequency combs[57,58], while microresonator-based frequency combs could provide 500 GHz repetition rates[59,60]. A 500 GHz repetition rate would result in a ≈3000-fold reduction in averaging time, or, for a given averaging time a $\sqrt{3000} = 55\times$ improvement in SNR. However, there are challenges to increasing the repetition rate while simultaneously maintaining the average

comb output power and spectral coverage. Nonetheless, recent reports of dark-pulse microresonators and dissipative Kerr soliton combs have shown >200 GHz repetition rate and on-chip powers exceeding 10 mW[61,62]. With further improvements in efficiency and output coupling, these comb sources could be well-suited for DCPAS.

If we assume a 500-GHz repetition rate, the same excitation power used here, and a tenfold stronger absorbance by illuminating in the visible or mid-infrared, the acquisition time is reduced by a factor of 300,000 so that a 2-h acquisition time drops to 24 milliseconds. Depending on the application, this could be further reduced by increasing the incident optical power. As shown in Fig. 5b, a linear increase in optical power leads to a linear improvement in SNR and to a corresponding quadratic decrease in the required averaging time. In some cases, additional considerations will limit the incident power for a given application. For example maximum permissible exposure limits for in vivo samples[63] and potential thermal damage for in vitro samples must be avoided.

In addition to increasing the incident optical power, the 24-millisecond acquisition time could also be decreased by decreasing the measurement noise. As the PA noise floor is limited by the transducer noise, improved transducer readout electronics or lower-noise transducing elements would directly improve the SNR[43] and lead to a quadratic reduction in acquisition time. In this initial demonstration of DCPAS, we did not choose an optimal transducer, but rather the 7.5 MHz transducer used is similar to transducers typically employed for conventional PAS experiments. The ability to adjust the multi-heterodyne frequencies of the optical excitation in DCPAS would allow the use of transducers with center frequencies and bandwidths both ranging from ≈1 MHz to ≈100 MHz.

The arguments mentioned above suggest that DCPAS could be useful for applications such as spectrally resolved medical imaging. In addition to considering these arguments, we also evaluated the potential of DCPAS by directly comparing DCPAS with a PAS system using a tunable pulsed optical parametric oscillator (OPO). As shown in Supplementary Fig. 1 and Supplementary Fig. 2, the current DCPAS implementation achieves a similar SNR to that obtained with pulsed PAS with 20 nJ pulse energies after accounting for wavelength multiplexing. Existing fiber-coupled OPO-based PAS systems can provide up to 20× more pulse energy and a correspondingly higher SNR. However, given the scaling arguments above, the use of higher repetition rate combs could improve DCPAS SNR by a factor of 55×. Therefore, an improved DCPAS implementation would be competitive with PAS systems that use tunable pulsed lasers while enabling many spectral elements to be acquired simultaneously without the extra time and potential systematics introduced by spectral scanning.

Finally, DCPAS would also benefit from a more accurate normalization procedure to remove systematic wavelength-dependent variations in the PA response. Our normalization procedure assumes that optical scattering, sample heating, pressure wave generation, and pressure wave propagation are independent of the multi-heterodyne frequencies in the optical intensity. Any frequency-dependent effects in these processes would skew the inferred absorption spectrum because of the coupling between ultrasonic and optical frequencies delineated by Eq. (2). One could address this problem by developing a model for the frequency dependence of PA signal generation under dual-comb excitation. This frequency dependence would be included in the $\beta$ proportionality factor in Eq. (4). Alternatively, one could use a reference sample with a known (and ideally flat) absorption spectrum to characterize the frequency response of the DCPAS system[23]. In applications where it is not necessary to determine quantitative absorption spectra, these issues might be completely

circumvented by applying principal component analysis to detect specific substances in DCPAS measurements[8]. Another limitation of the normalization procedure used here is that the optical excitation spectrum is only measured before the light hits the sample. This could be improved by using optical inversion calculations to obtain the optical spectrum as a function of depth. In that case, one could use a depth-resolved version of Eq. (4) to normalize the PA response[64].

Overall, the results demonstrated here show that dual-comb excitation can generate PA signals. These PA signals, in turn, reflect the absorption spectra of sample materials and can be used to detect and identify different organic solids. Furthermore, we propose several strategies to increase DCPAS acquisition speeds and to develop more accurate normalization procedures. With these improvements, DCPAS could become a useful new technology for high-speed, label-free spectroscopy and possibly imaging. The authors would like to acknowledge a recent preprint demonstrating DCPAS for detection of gaseous acetylene[65]. This shows yet another potential application of DCPAS.

## Methods

**Samples**. We used three different samples for DCPAS in this work. One sample consists of a ≈1 mm × 2 mm film of vertically aligned carbon nanotubes (VACNTs) embedded in an ≈240 μm thick film of poly(dimethylsiloxane) (PDMS). The VACNTs are multi-walled CNTs with an estimated tube diameter of ≈10 nm to ≈20 nm based on measurements of similarly prepared samples. The thickness of the VACNT layer varies from ≈15 μm to ≈40 μm (see Supplementary Fig. 4). The PDMS film is ≈2.5 mm wide and ≈5 mm long, with several millimeter-wide regions that contain no VACNTs (see Supplementary Fig. 5). One of these regions is used as a second PA sample that consists of only PDMS. The third sample is a piece of paraffin-based paper (Parafilm "M", ≈120 μm thick) cut into a ≈5 mm × 5 mm square (the use of trade names is necessary to specify the experimental results and cannot imply endorsement by the National Institute of Standards and Technology.). Each of these samples is laminated onto the top of a N-BK7 wedge by gently applying pressure with a pair of tweezers. The wedge is sealed against an opening in the bottom of a water bath so that the sample is immersed in the water (location (v) in Fig. 2) and oriented toward the transducer. The distance from the sample to the transducer is equal to the transducer focal length of approximately 2 cm. We also measured the optical absorption spectra of the PDMS and paraffin in transmission-mode with a NIR spectrophotometer (Perkin-Elmer Lambda 1050 equipped with a 3-detector module [The use of trade names is necessary to specify the experimental results and cannot imply endorsement by the National Institute of Standards and Technology]) for comparison with the spectra obtained in the DCPAS system. The same PDMS sample was used for DCPAS and spectrophotometer measurements. The spectrophotometer illuminated a portion of the PDMS sample that contained no VACNTs; however, this illumination was not in the exact same location as the dual-comb excitation. The paraffin sample used for DCPAS could not be used for spectrophotometer measurements because it displayed too much optical scattering for transmission measurements. For transmission measurements, we reduced the optical scattering by placing the paraffin between two glass microscope slides, heating the glass to 70° C, and applying pressure. Spectrophotometer measurements of this sample were made after allowing it to cool to room temperature.

**DCPAS system design**. Our DCPAS system uses two self-referenced fiber frequency combs with repetition rates, $f_{rep}$, of ≈160 MHz, described in more detail in refs. [44] and [66]. Each comb emits ≈40 mW of light centered at 192 THz (1560 nm) with a 3-dB bandwidth of ≈45 nm. This output is filtered with a 1550 nm ± 5 nm bandpass filter and then amplified to ≈400 mW with a triple-pumped erbium-doped fiber amplifier. After amplification, the light from each comb is spectrally broadened in ≈3 cm of highly nonlinear fiber with a dispersion of ≈2.2 ps nm$^{-1}$ km$^{-1}$. This yields ≈60 mW of light from each comb spanning 158 THz to 187 THz (location i in Fig. 2). This 158 THz to 187 THz light from each comb is combined on a beam-splitter, transmitted through a 187 THz low-pass filter and a linear polarizer, and coupled into single-mode PM1550 fiber (location ii in Fig. 2) for transport to the PA setup. The light is then launched into free space by a $f$ = 50.8 mm reflective collimator. About 88% of the light is transmitted through a CaF$_2$ wedge and ≈6% is reflected by both the front and back faces. The back reflection is coupled into single-mode fiber and detected by an extended InGaAs photodiode (location (iii) in Fig. 2) to record the optical excitation spectrum. The front reflection is coupled into a single-mode fiber delay line, for ≈100 μs of delay. Then it is detected by a standard InGaAs photodiode to provide a trigger for the FPGA data acquisition system.

The ≈25 mW of light transmitted through the CaF$_2$ wedge is focused by a $f$ = 45 mm achromatic lens to a spot size of 4 μm on the sample to generate the PA

signal. Absorption by the sample generates pressure waves that are detected by a polytetrafluoride ultrasound transducer with a frequency response that is peaked at 7.5 MHz with a ≈7.5 MHz bandwidth (Supplementary Note 2) and a focal length of 2 cm (Olympus Panametrics A320S [the use of trade names is necessary to specify the experimental results and cannot imply endorsement by the National Institute of Standards and Technology]). This transducer is mounted on a 3-D translation stage and positioned to maximize the PA signal, which occurs when the transducer focus overlaps the optical focus of the incident comb light. In addition, the sample is translated vertically to place it at the common focal plane of the comb light and transducer. As there is 2 cm (transducer focal length) of water between the sample and the transducer and because the absorption coefficient of water is $\mu_A$ > 6.5 cm$^{-1}$ for the optical frequencies in this experiment, no light reaches the face of the transducer. We have also verified that when the sample is translated out of the common focus of the illumination and transducer, no signal is measured.

After the ultrasound transducer, the PA signal is amplified by ≈80 dB and filtered with a 15 MHz low-pass filter before it is input to the data acquisition system, as shown in Fig. 2. The optical excitation signal is filtered with a 50 MHz low-pass filter after the photodetector and before the data acquisition system. The PA signal and the optical excitation signal are digitized on separate channels of the data acquisition system, and the resulting time-domain signals are interferograms, as shown in Fig. 3a, b. For the SNR calculations shown in Fig. 5, the time-domain SNR is defined as the peak height of the interferogram divided by the standard deviation of the noise outside of the interferogram centerburst (see Supplementary Fig. 2 and Supplementary Note 1). As the SNR is low for the weakly absorbing polymer samples, multiple interferograms are coherently averaged in real-time as is common in DCS[34,45]. We implement this signal averaging on a data acquisition system that consists of a multi-channel 14-bit analog-digital converter and a field-programmable gate array (FPGA). The firmware in the FPGA removes residual phase noise from the interferograms before co-adding them for long-term averaging[34,45]. To measure slow phase drifts, the phase correction firmware requires a stronger signal than is present in a single PA interferogram. We resolve this issue by adding the optical DCS voltage to the DCPAS voltage as a delayed trigger signal as shown in Fig. 2.

## Data availability
Data and processing scripts are available on request from the authors.

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

## Acknowledgements

We would like to acknowledge G. Ycas for fruitful discussion and development of FPGA code. We acknowledge helpful comments from E.M. Waxman and A.S. Kowligy. J.T.F. and A.M.G. both conducted their research while the authors held NRC Research Associateship awards at NIST.

## Author contributions

J.T.F helped design and build the DCPAS system. He also acquired and analyzed the data and wrote the manuscript. E.B., G.M.C., D.H., E.V.H., E.F.P., and helped design and build the DCPAS system. E.B., K.A.B., F.R.G., J.H., N.R.N., I.C., and K.C.C. provided guidance for experimental design and interpretation of data. F.R.G. and K.C.C. helped with data analysis. C.S.Y. and J.H. fabricated the PDMS/VACNT sample. A.M.G., K.A.B., and J.T.F. performed the PAS measurements with the optical parametric oscillator. All authors approve of the submitted version of the manuscript and agree to be personally accountable for his/her own contributions and to ensure that questions related to the accuracy or integrity of any part of the work, even ones in which he/she was not personally involved, are appropriately investigated, resolved, and the resolution documented in the literature.

## Competing interests

The authors declare no competing interests.

**Additional information**

