## [Peer Review File · Nature Communications]

Reviewers' comments:

Reviewer #1 (Remarks to the Author):

The manuscript of J.T. Friedlein reports the first experimental demonstration of a new method called Dual-comb photoacoustic spectroscopy (DCPAS). The work is clearly novel, and the manuscript is carefully prepared and well organized.

As the authors point out, the DCPAS method offers potential advantages in chemically specific label-free imaging, for example in life sciences. The paper thus has potential for high and multidisciplinary impact. However, the potential impact of the work critically depends on whether the DCPAS method can be used to increase measurement speed in spectroscopic photoacoustic imaging. This possibility cannot be properly estimated based on the information given in the current version of the manuscript. Before making my final decision on the manuscript, I would like to see this point addressed by the authors – please see the comments below for details.

I have just one major comment, but an important one:

The paper should include a comparison of the achieved detection sensitivity against a typical detection sensitivity obtained and/or needed with the existing methods and applications. I think such comparison is essential in order to put the work into context and, in particular, to estimate the feasibility of the new method in the intended applications. It is understandable that a proof-of-concept demonstration is far from optimum and further improvements are needed. Some of the needed improvements are already nicely discussed at the end of the manuscript, and the addition of sensitivity comparison would thus make it possible to judge the potential impact of the work.

Ideally, the sensitivity comparison should be based on a direct comparison of the DCPAS measurements against measurements of the same samples with an existing photoacoustic imaging instrument. Alternatively, a comparison with values given in literature could be used – although such comparison is somewhat problematic due to different measurement conditions, ultrasonic transducers, etc. One option would be to compare the minimum detectable absorbance of DCPAS with minimum detectable absorption calculated from the measurements of ref 6, where glucose was detected using the same wavelength region as in the DCPAS demonstration. (Admittedly, this is not a fair comparison, owing to a more complex sample of [6]. However, it would allow one to get an order of magnitude estimate of the detection limit for a certain averaging time).

Minor comments:

As for possible further improvements, the laser frequency comb technology is discussed in detail but the acoustic/ultrasonic transducer part is mentioned just briefly. For a more balanced discussion, it would be useful to have more information about the transducer and its performance. Why was this particular transducer chosen for the experiments? Is a transducer that works well in a conventional PA imaging system also optimal for DCPAS? In gas-phase PAS, the sensitivity of acoustic sensor can be extremely high owing to low acoustic frequencies. In PA imaging, high frequencies are needed, which reduces the detection sensitivity but offers more bandwidth for dual-comb detection. This latter point is briefly mentioned in Methods section, but it would be interesting to know more about the design considerations and if the transducer has room for further improvements.

Considering the modest performance (power level) of today's microresonator-based frequency combs, I find the 500 GHz comb discussion quite speculative. (On the other hand, the dark-pulse microresonator combs may provide a solution to this issue). In order to better estimate the possible

use of microresonator-based combs in DCPAS imaging, it would be important to know what kind of power levels are actually needed (see the major comment).

Introduction: comb teeth widths could perhaps be given in frequency units or in spectroscopic wavenumber units. (Frequency units have been used elsewhere in the manuscript, and would be a logical choice).

Reviewer #2 (Remarks to the Author):

This manuscript demonstrated for the first time photoacoustic spectroscopy with dual frequency combs. Although pulsed laser-based photoacoustics spectroscopy has been performed in solids and liquids for decades, and recently comb photoacoustic spectroscopy has been performed with a single frequency comb, the result the authors present is intriguing and very interesting.

The authors should discuss more convincingly, what the advantage is of using this approach instead of using more traditional pulsed laser photoacoustics or Fourier Transform Photoacoustics. This new type of research is challenging, but must be convincingly described. The number and type of experiments that demonstrate the method could be improved and better described/explained.

From the photoacoustic detection of the absorption in the thin layer there are a number of points to address:

- The thickness of the thin layer thickness is not given. As such, neither absorption coefficients/sensitivities are calculated nor comparisons are made with pulsed laser photoacoustic experiments.
- The nanotubes are not described (1x10 mm film, 3rd dimension?). Size of the nanotubes? Nanotubes are embedded in PDMS (400 micron thick), but how are they distributed in this film? How is this configuration exactly, compare to the 4 micron light spot?
- Spectra of carbon nanotubes are shown embedded in PDMS, next to spectra of PDMS itself. Since both responses are in arbitrary units, but can they be compared to each other?
- The generated acoustic signal is compared with the absorbance after normalizing it on the wavelength dependent power of the light. As discussed in Fig. 3C and 3D, there is quite some difference in acoustic response at low frequencies, which is caused by the response of the transducer at low frequencies (peak at 7.5 MHz, bandwidth 7.5 MHz). For this, a datasheet response of the transducer should be useful.
- At higher frequencies, the detector response seems also not optimal, is this due to the acoustic response of the transducer, or is this due to internal energy redistribution at molecular level within the material, generating not a direct acoustic response (see for example the peaks at 1680-1700 in paraffin)?
- The thin layer is optically thin and it seems that quite some light will hit the detector through the water layer, please comment on this.
- In the discussion, the authors argue not to go to the mid-infrared for biological material, but other researchers show that this can be well achieved in thin layers.
- Furthermore, the authors discuss how to improve the method by going to larger comb spacing up to 500 GHz. They oversee the problem of acoustic detection, are there sensitive acoustic transducers at these frequencies?

There are a number of small remarks:

- The first time μ is mentioned in the formula of the photoacoustic pressure wave, it is stated as an

absorption coefficient, it should however be absorption.

-For a spectroscopist it is not correct to mention energies in wavelength and bandwidth in nanometers. The wavelength of light depends on the material in which it propagates, the light frequency not. So I suggest change/add this.

- The abbreviation PDMS is not explained the first time it appears in the manuscript.

- Fig. 4 demonstrates the linearity of the optical power with the time dependent SNR of the photoacoustic signal. The latter is unclearly defined: a time dependent acoustic transient, such as Fig 2C? How is the SNR calculated from this?

- From Fig. 4 it is clearly seen that there is no saturation ongoing for the nanotubes, is this the same for the other materials used?

- It is unclear why Figure S1 is supplementary information, this figure is needed to understand what I aimed for in the text.

Reviewer #3 (Remarks to the Author):

In this manuscript, the authors demonstrate the first dual-comb photoacoustic measurement. Photoacoustic detection is widely used in sensitive spectroscopy of both gas-phase and condensed-phase samples. Photoacoustic spectroscopy has proved useful in spectral imaging systems, especially with samples that transmit poorly the optical wavelengths of interest. Dual-comb spectroscopy is an emerging technique to disperse the spectrum of a frequency comb, without a need for a separate spectrometer. This allows fast spectrum acquisition without mechanical limitations from a spectrometer, but often imposes heavy requirements to the stability of the light source itself. Additionally, dual-comb spectroscopy generally requires high-bandwidth detectors, if broad spectra are desired, which can be a challenge with acoustic detection. In this manuscript, the authors use a relatively high-bandwidth ultrasound detector, together with highly stabilized frequency combs, which can efficiently take advantage of the limited bandwidth of an acoustic detector.

My only concern with the manuscript is that the demonstrated measurement system has a rather low signal-to-noise ratio with reasonable averaging times, and it capitalizes poorly on the potential benefits of using dual-comb spectroscopy in comparison to, for example, a more traditional FTS measurement with a single comb. This is mainly due to using combs with low repetition rate in comparison to the required resolution. The high bandwidth acoustic detector also generally comes at a cost of sensitivity. The disadvantages have been partially counteracted by apodizing the interferograms, but in doing so, a large portion of the measurement time is effectively wasted. However, the authors clearly state that the reported measurements are a proof of concept, and the manuscript lays out practical steps required to improve the measurement and to properly reach the advantages offered by dual-comb spectroscopy. This mainly comes down to using combs with more appropriate repetition rate and wavelength. Such combs are less widely available and still come with plenty of challenges, but this demonstration of dual-comb photoacoustic spectroscopy would further motivate their development and optimization.

This demonstration that dual-comb technique can be applied with photoacoustic spectroscopy would be of high interests to a relatively wide reader base, including researchers in the fields of dual-comb spectroscopy, photoacoustic spectroscopy or spectral imaging, as well as researchers working to produce novel frequency combs, such as those with high repetition rates. The measurements and the results are described clearly and in high technical detail. In terms of the presentation I thought I should mention that when the manuscript was printed out, it appears that some parts of the figure caption texts appear with clearly varying shades of blue and the light grey trace used in figure 3A and

B is practically invisible. All in all, the results are novel and convincing, and of high enough impact that I would support the publication of the manuscript.

To all reviewers: Thank you for taking the time to carefully consider our manuscript. We have addressed your concerns in the document below and appreciate that addressing them has improved the manuscript and significantly increased the impact of our work. Below we list how we have addressed each reviewer's comments.

Reviewer 1: *The manuscript of J.T. Friedlein reports the first experimental demonstration of a new method called Dual-comb photoacoustic spectroscopy (DCPAS). The work is clearly novel, and the manuscript is carefully prepared and well organized.*

As the authors point out, the DCPAS method offers potential advantages in chemically specific label-free imaging, for example in life sciences. The paper thus has potential for high and multidisciplinary impact. However, the potential impact of the work critically depends on whether the DCPAS method can be used to increase measurement speed in spectroscopic photoacoustic imaging. This possibility cannot be properly estimated based on the information given in the current version of the manuscript. Before making my final decision on the manuscript, I would like to see this point addressed by the authors – please see the comments below for details.

I have just one major comment, but an important one:

The paper should include a comparison of the achieved detection sensitivity against a typical detection sensitivity obtained and/or needed with the existing methods and applications. I think such comparison is essential in order to put the work into context and, in particular, to estimate the feasibility of the new method in the intended applications. It is understandable that a proof-of-concept demonstration is far from optimum and further improvements are needed. Some of the needed improvements are already nicely discussed at the end of the manuscript, and the addition of sensitivity comparison would thus make it possible to judge the potential impact of the work.

Ideally, the sensitivity comparison should be based on a direct comparison of the DCPAS measurements against measurements of the same samples with an existing photoacoustic imaging instrument. Alternatively, a comparison with values given in literature could be used – although such comparison is somewhat problematic due to different measurement conditions, ultrasonic transducers, etc.

We agree with the reviewer's general comment, and we agree that a comparison with the literature is problematic because of the different measurement conditions. Because of those challenges, we made a series of new measurements allowing an extensive comparison to a traditional approach as requested by the reviewer. Specifically, we replaced the dual-comb source in our setup with a tripled YAG-pumped optical parametric oscillator (OPO) that could produce 4-ns pulses at varying wavelengths. We then compared the SNR in both the time-domain and spectral domain between this OPO-PAS system and DCPAS.

The results of this comparison are described in detail in Section 1 of the Supplementary Information (and not repeated here in interest of length). In the main body of the text we summarize the results of this comparison by noting, on page 12, that:

“The arguments mentioned above suggest that DCPAS could be useful for applications such as spectrally resolved medical imaging. In addition to considering these arguments, we also evaluated the potential of

DCPAS by directly comparing DCPAS with a PAS system using a tunable pulsed optical parametric oscillator (OPO). As shown in Figure S1-S2, the current DCPAS implementation achieves a similar SNR to that obtained with pulsed PAS with 20 nJ pulse energies after accounting for wavelength multiplexing. Existing fiber-coupled OPO-based PAS systems can provide up to $50 \times$ more pulse energy and a correspondingly higher SNR. However, given the scaling arguments above, the use of higher repetition rate combs could improve DCPAS SNR by a factor of $55 \times$. Therefore, an improved DCPAS implementation would be competitive with PAS systems that use tunable pulsed lasers while enabling many spectral elements to be acquired simultaneously without the extra time and potential systematics introduced by spectral scanning.”

Reviewer 1: As for possible further improvements, the laser frequency comb technology is discussed in detail but the acoustic/ultrasonic transducer part is mentioned just briefly. For a more balanced discussion, it would be useful to have more information about the transducer and its performance. Why was this particular transducer chosen for the experiments? Is a transducer that works well in a conventional PA imaging system also optimal for DCPAS? In gas-phase PAS, the sensitivity of acoustic sensor can be extremely high owing to low acoustic frequencies. In PA imaging, high frequencies are needed, which reduces the detection sensitivity but offers more bandwidth for dual-comb detection. This latter point is briefly mentioned in Methods section, but it would be interesting to know more about the design considerations and if the transducer has room for further improvements.

We selected this particular off-the-shelf transducer because of its availability. The transducer response is provided in the new Figure S3 in the supplementary information section reproduced here:

Figure S3. Transducer responsivity (blue dotted line) compared to the multiheterodyne frequencies generated by the dual-comb optical excitation (black solid line). Note that the agreement is not fortuitous. Rather, the relative repetition and offset frequencies of the two frequency combs are selected so that the multiheterodyne spectrum is matched to the transducer response. A transducer with a different bandwidth or center frequency could be accommodated by adjusting the frequency comb parameters. For instance, if the transducer spectrum narrows, one can similarly narrow the multiheterodyne spectral bandwidth by reducing the difference in repetition frequencies, assuming the two combs are sufficiently phase coherent.

As noted in the caption of Fig. S3, the center frequency and bandwidth of the transducer is not critical because the spectrum of the multi-heterodyne signal is highly flexible. Principles for transducer selection in optical resolution photoacoustic microscopy (OR-PAM) should also apply to DCPAS. We have added this detail in the discussion section (page 12) of the manuscript as follows:

“In addition to increasing the incident optical power, the 24-millisecond acquisition time could also be decreased by decreasing the measurement noise. Because the PA noise floor is limited by the transducer noise, improved transducer readout electronics or lower-noise transducing elements would directly improve the SNR [43] and lead to a quadratic reduction in acquisition time. In this initial demonstration of DCPAS, we did not choose an optimal transducer, but rather the 7.5 MHz transducer used is similar to transducers typically employed for conventional PAS experiments. The ability to adjust the multiheterodyne frequencies of the optical excitation in DCPAS would allow the use of transducers with center frequencies and bandwidths both ranging from ≈ 1 MHz to ≈ 100 MHz.”

We also moved the paragraph that discusses transducer bandwidth from the Methods section to the introduction. Within this paragraph, we’ve clarified that we can map the dual-comb excitation to a narrow transducer bandwidth if necessary.

“One advantage of DCPAS is that the bandwidth and center frequency of the generated acoustic signal, corresponding to the amplitude modulation on the optical signal, can be adjusted independently of the optical bandwidth and center frequency. This adjustment is made in a controlled fashion by tuning the

frequency locking of the two combs to adjust $\nu_{2,0}$, $\nu_{1,0}$, and Δf_{rep} , and therefore, the multiheterodyne frequencies f_i . This allows any optical spectrum to generate a PA signal that can be detected by an ultrasound transducer despite the transducer's limited bandwidth. In this demonstration, we use lasers with ≈ 160 MHz repetition rates and set the difference in repetition rates (tooth spacings), $\Delta f_{rep} = f_{rep,2} - f_{rep,1}$, to be 66.81 Hz. This allows us to map the roughly 15 THz optical bandwidth (full width at -10 dB) into an acoustic bandwidth of $(15 \text{ THz}) \times (\Delta f_{rep}/f_{rep}) \approx 6.3 \text{ MHz}$ [32] while centering this modulation signal at the peak of the transducer responsivity, 7.5 MHz. In order to maintain this mapping as well as the optical coherence required for long term averaging of the signals, both combs are fully stabilized in the manner described in ref. [44].”

Reviewer 1: *Considering the modest performance (power level) of today's microresonator-based frequency combs, I find the 500 GHz comb discussion quite speculative. (On the other hand, the dark-pulse microresonator combs may provide a solution to this issue). In order to better estimate the possible use of microresonator-based combs in DCPAS imaging, it would be important to know what kind of power levels are actually needed (see the major comment).*

We agree with the reviewer's concern, and perhaps we are overly optimistic. We have added references to dark-pulse microresonator combs and dissipative Kerr soliton combs. With improvements in efficiency and output coupling, these lasers should enable 500-GHz DCPAS at powers exceeding 10 mW. If 500-GHz combs could provide 25 mW of average power, then, as noted in the discussion (page 11), for a fixed SNR, the same spectra could be measured in 1/3000 the time used in our experiments. That means that the spectra displayed as solid black curves in Figure 4 could be captured in 2.4 s. The new text (page 11) addresses the possibilities of dark-pulse and dissipative Kerr soliton combs:

“However, there are challenges to increasing the repetition rate while simultaneously maintaining the average comb output power and spectral coverage. Nonetheless, recent reports of dark-pulse microresonators and dissipative Kerr soliton combs have shown >200 GHz repetition rate and on-chip powers exceeding 10 mW [61,62]. With further improvements in efficiency and output coupling, these comb sources could be well-suited for DCPAS.”

Reviewer 1: *Introduction: comb teeth widths could perhaps be given in frequency units or in spectroscopic wavenumber units. (Frequency units have been used elsewhere in the manuscript, and would be a logical choice).*

We have changed wavelength units to frequency units in all cases.

Response to Reviewer 2

Reviewer 2: *This manuscript demonstrated for the first time photoacoustic spectroscopy with dual frequency combs. Although pulsed laser-based photoacoustics spectroscopy has been performed in solids and liquids for decades, and recently comb photoacoustic spectroscopy has been performed with a single frequency comb, the result the authors present is intriguing and very interesting.*

The authors should discuss more convincingly, what the advantage is of using this approach instead of using more traditional pulsed laser photoacoustics or Fourier Transform Photoacoustics. This new type of research is challenging, but must be convincingly described. The number and type of experiments that demonstrate the method could be improved and better described/explained.

We have clarified in the introduction that the main advantage of DCPAS over pulsed PAS is the fact that an entire spectrum is obtained in a single interferogram. Compared to Fourier Transform Photoacoustics,

DCPAS has the potential to be more compact and possibly more robust since there is no mechanical delay line. Moreover, thermal sources for FTIR have low power, which can limit the measurement speed, while supercontinuum sources can have significant laser intensity noise, which can limit the SNR. However, we agree that much more data will be needed to clarify the true comparison between these two closely related methods. We have reworded the introduction (page 2) to clarify these points as:

“In vivo and in vitro medical imaging and spectroscopy face the fundamental challenge of strong optical scattering in biological tissues. This challenge has led researchers to use the photoacoustic (PA) effect, where localized light absorption generates acoustic waves that can be detected from deep within tissue [1–4]. PA measurements have an advantage over pure optical methods because light only has to travel through the sample to the absorber to create an acoustic wave, but the light does not need to travel back out of the sample to a detector. Other advantages of PA measurements are that scattered optical illumination can still induce a PA signal, and the generated acoustic waves are only weakly scattered by tissue [5]. In addition, by measuring the PA response versus excitation wavelength, it is possible to identify and quantify substances based on their unique absorption features. For example, researchers have used this method, called PA spectroscopy, to quantify glucose levels *in vivo* [6–8]. A more advanced technique for medical diagnostics and functional imaging, called multispectral optoacoustic tomography (MSOT) or photoacoustic tomography (PAT), has also been demonstrated. This technique provides images of multiple exogenous and endogenous species at multiple wavelengths and uses spectral unmixing algorithms to determine the spatial distribution of each species [2,9,10]. This technique has been used to differentiate adipose tissue from cholesterol [11,12], measure intramuscular fat distributions [13], map tumor size and shape [14,15], identify white matter loss in spinal cord injuries [16], quantify blood oxygen saturation [17,18], and even to construct a label-free molecular map of a mouse fibroblast cell [19].

One challenge of PA spectroscopy, or PAS, measurements is that they require a broadband optical excitation source with a spectrally resolved detection or illumination method. Traditionally, PA researchers have used optical parametric oscillators or other tunable lasers, which require sequential image acquisition over narrow optical frequency bands. Although it is possible in simple ratiometric PAS to use only a couple of optical frequency bands, measuring in more bands allows the identification of additional species and improves detection sensitivity [20]. However, sequentially scanning the optical frequency can be time consuming and can lead to potential issues with sample drift or damage between images at different optical frequencies [9,21]. Broadband thermal sources using a Fourier transform spectrometer (FTS) [22–25] for optical frequency resolution have been used to overcome sequential scanning, but the low optical power and slow scanning delay arm have limited the application of these systems. Alternatively, broadband supercontinuum coherent laser sources may provide higher power than a thermal source, but still require either sequential frequency scanning or physical scanning of an FTS delay arm and can suffer from high laser intensity noise. Optical frequency combs [26–28] are an alternative light source for broadband PAS and multispectral PAT because they simultaneously generate thousands of discrete optical frequency bands. These frequency bands are called comb teeth because, like the teeth of a comb, they are evenly spaced and very narrow, with absolute linewidths below 120 kHz [29]. The comb teeth of a frequency comb are temporally and spatially coherent and propagate in a single spatial mode with high brightness. Recent work has shown that frequency combs are a promising light source for PAS. Researchers have demonstrated highly sensitive and precise measurements of methane absorption spectra with frequency-comb-based PAS [30,31]. These demonstrations relied on a FTS for spectral selectivity and thus required physical scanning of the optical pathlength. It is possible, however, to eliminate the need for the FTS altogether by using a second comb like a local oscillator. By exploiting the coherence of optical frequency combs, one can capture an entire absorption spectrum on a

single photodetector without wavelength or delay arm scanning, as is done in dual-comb spectroscopy (DCS) [32]. DCS provides broad spectral coverage, high sensitivity, and high spectral resolution for identification of multiple chemical species [33–37]. It also can provide rapid spectral measurements for monitoring dynamic biological processes [38] as well as for coherent Raman imaging [39,40]. These advantages should be applicable to PAS using a technique analogous to DCS. This new method of dual-comb PAS (DCPAS), which was suggested in [31], can be used across broad spectral windows without sequential wavelength scanning and can be implemented with a compact, stable, and robust dual-comb spectrometer.”

Also, in response to reviewer 1, we made an experimental comparison of DCPAS with traditional pulsed laser photoacoustics. This is discussed in the response above and in section 1 of the supplementary information. That discussion addresses the potential of DCPAS to operate as fast as pulsed PAS without sequential wavelength scanning.

Reviewer 2: From the photoacoustic detection of the absorption in the thin layer there are a number of points to address: The thickness of the thin layer thickness is not given. As such, neither absorption coefficients/ sensitivities are calculated nor comparisons are made with pulsed laser photoacoustic experiments.

We agree that a more detailed description of the samples was needed. In the Methods section (page 13), we note the PDMS, paraffin and VACNT thicknesses. The modified text reads:

“We used three different samples for DCPAS in this work. One sample consists of a $\approx 1 \text{ mm} \times 2 \text{ mm}$ film of vertically aligned carbon nanotubes (VACNTs) embedded in an $\approx 240 \text{ }\mu\text{m}$ thick film of poly(dimethylsiloxane) (PDMS). The VACNTs are multi-walled CNTs with an estimated tube diameter of $\approx 10 \text{ nm}$ to $\approx 20 \text{ nm}$ based on measurements of similarly prepared samples. The thickness of the VACNT layer varies from $\approx 15 \text{ }\mu\text{m}$ to $\approx 40 \text{ }\mu\text{m}$ (see Figure S4 in the supplementary information). The PDMS film is $\approx 2.5 \text{ mm}$ wide and $\approx 5 \text{ mm}$ long, with several millimeter-wide regions that contain no VACNTs (see Figure S5 in the supplementary information). One of these regions is used as a second PA sample that consists of only PDMS. The third sample is a piece of paraffin-based paper (Parafilm “M”, $\approx 120 \text{ }\mu\text{m}$ thick) cut into a $\approx 5 \text{ mm} \times 5 \text{ mm}$ square [65].”

For comparison to the spectrophotometer measurements, we note (page 13):

“The same PDMS sample was used for DCPAS and spectrophotometer measurements. The spectrophotometer illuminated a portion of the PDMS sample that contained no VACNTs; however, this illumination was not in the exact same location as the dual-comb excitation. The paraffin sample used for DCPAS could not be used for spectrophotometer measurements because it displayed too much optical scattering for transmission measurements. For transmission measurements, we reduced the optical scattering by placing the paraffin between two glass microscope slides, heating the glass to 70 C , and applying pressure. Spectrophotometer measurements of this sample were made after allowing it to cool to room temperature.”

We also provided a new section (section 2) of the supplementary which discusses the samples more thoroughly and includes photos of the PDMS and VACNT samples. We do not copy that entire section here, but we note that it includes an absorbance estimate for the VACNTs.

“We measured the total absorption of the VACNTs in PDMS and found that, near the edge of the sample $\approx 5\%$ of the incident light was transmitted through the sample and near the center of the sample $\approx 15\%$ of the light was transmitted.”

Finally, instead of quoting a sensitivity in absorbance or thickness units we opted for a comparison to traditional pulsed PA spectroscopy as mentioned in the response to the previous comment and discussed in detail in the response to Reviewer 1.

Reviewer 2: The nanotubes are not described (1x10 mm film, 3rd dimension?). Size of the nanotubes? Nanotubes are embedded in PDMS (400 micron thick), but how are they distributed in this film? How is this configuration exactly, compare to the 4 micron light spot?

The reviewer highlights several important details that were not clarified in the initial submission. We have included these details in the Methods section (page 13) and section 2 of the supplementary information.

In the Methods (page 12-13) section, the modified text reads:

“The VACNTs are multi-walled CNTs with an estimated tube diameter of ≈ 10 nm to ≈ 20 nm based on measurements of similarly prepared samples. The thickness of the VACNT layer varies from ≈ 15 μm to ≈ 40 μm (see Figure S4 in the supplementary information). The PDMS film is ≈ 2.5 mm wide and ≈ 5 mm long, with several millimeter-wide regions that contain no VACNTs (see Figure S5 in the supplementary information). One of these regions is used as a second PA sample that consists of only PDMS.”

Below is Figure S5 from the supplementary information. This figure shows that the optical excitation spot size is much smaller than the lateral dimensions of the VACNT and PDMS samples.

Figure S5. Top-down image of the VACNTs and PDMS. For measurements of VACNTs, the illumination was contained entirely in the region covered by VACNTs. For measurements of PDMS, the illumination was contained entirely in the region with no VACNTs.

Reviewer 2: “Spectra of the carbon nanotubes are shown embedded in PDMS, next to spectra of PDMS itself. Since both responses are in arbitrary units, but can they be compared to each other.”

We have replaced the a.u. scaling in Figures 3, 4A, and 4B with measured voltages. Therefore, the magnitudes shown in these figures for the different samples can be directly compared to each other. We have also clarified the ability to directly compare these data in the discussion of Figure 4 (page 8 of main text):

“Because the polymers absorb more weakly than the VACNTs and are less efficient at generating acoustic waves, they produce a DCPAS signal that is $50 \times$ and $100 \times$ weaker than the VACNTs, as can be seen from comparing Figure 3C to Figure 4A and 4B, respectively.”

Reviewer 2: *“The generated acoustic signal is compared with the absorbance after normalizing it on the wavelength dependent power of the light. As discussed in Fig. 3C and 3D, there is quite some difference in acoustic response at low frequencies, which is caused by the response of the transducer at low frequencies (peak at 7.5 MHz, bandwidth 7.5 MHz). For this, a datasheet response of the transducer should be useful.”*

We acknowledge that the transducer responsivity is helpful information to understand the results reported in our manuscript. We have included the transducer responsivity in the supplementary information Figure S3 and have referenced this in the main text (page 5):

“The PA signals generated by the samples then propagate through a water bath to a polytetrafluoride ultrasound transducer with a frequency response that is peaked at 7.5 MHz with an ≈ 7.5 MHz bandwidth (see Figure S3 in the supplementary information).”

Reviewer 2: *“At higher frequencies, the detector response seems also not optimal, is this due to the acoustic response of the transducer, or is this due to internal energy redistribution at molecular level within the material, generating not a direct acoustic response (see for example the peaks at 1680-1700 in paraffin)?”*

As shown in the transducer datasheet mentioned above (Figure S3), the transducer response decreases at high frequencies. Because the peaks in the paraffin spectrum between 1680 nm and 1700 nm appear both in the DCPAS response and in the spectrophotometer measurements (see Fig. 4D of manuscript), we conclude that they are probably actual absorption features and not an artefact of the DCPAS measurement method. We acknowledge that these features are more pronounced in the normalized PA response (Fig. 4D, black curve) than in the spectrophotometer absorbance (Fig. 4D, dashed red curve), but we do not know why. One possibility is that the transducer responsivity is underestimated at high frequencies in the datasheet. Such an underestimate would increase the normalized PA response, \widehat{V}_{PA} , at high frequencies because \widehat{V}_{PA} is obtained by dividing the measured PA response by the transducer responsivity.

Reviewer 2: *“The thin layer is optically thin and it seems that quite some light will hit the detector through the water layer, please comment on this.”*

During our measurements, we verified that no light hits the transducer. We added this detail in the Methods section (page 13):

“Because there is 2 cm (transducer focal length) of water between the sample and the transducer and because the absorption coefficient of water is $\mu_A > 6.5 \text{ cm}^{-1}$ for the optical frequencies in this experiment, no light reaches the face of the transducer. We have also verified that when the sample is translated out of the common focus of the illumination and transducer, no signal is measured.”

Reviewer 2: *“The authors argue not to go to the mid-infrared for biological material, but other researchers show that this can be well achieved in thin layers.”*

We agree with the reviewer that DCPAS in the mid-IR might be useful for thin samples. We have reworded our discussion of that topic on page 11:

“However, mid-infrared frequency combs [34,35,38,52] could measure much stronger fundamental C-H rovibrational transitions. These stronger absorption features (up to several orders of magnitude) could provide a proportional increase in the PA signal strength and a corresponding quadratic decrease in the

required averaging time. However, strong water absorption would limit mid-infrared DCPAS to $< 100 \mu\text{m}$ penetration depths in biological tissue because the $1/e$ penetration depth is only $32 \mu\text{m}$ for excitation at $\nu_{\text{optical}} = 60 \text{ THz}$ [53]. Nevertheless, mid-infrared DCPAS could be useful for measuring thin samples in the same way that pulsed PAS can measure thin samples at mid-infrared wavelengths [8]. Moving to visible/near-infrared wavelengths would enable measurements of oxygenated and deoxygenated hemoglobin. This visible/near-infrared hemoglobin absorption is about 10 times stronger than lipid absorption at around 173 THz without any competing absorption from water [4]. Additionally, other exogenous agents and dyes with strong absorption features could be measured with visible/near-infrared wavelengths [9].”

Reviewer 2: “Furthermore, the authors discuss how to improve the method by going to larger comb spacing up to 500 GHz. They oversee the problem of acoustic detection, are there sensitive acoustic transducers at these frequencies?”

This question highlights an important point that could confuse future readers. The PA pressure wave generated by dual-comb illumination is modulated at the mixed-down multiheterodyne frequencies, not the repetition rates of the frequency combs. Therefore, one could use 500-GHz frequency combs without requiring a 500-GHz transducer just as we have used 160 MHz frequency combs with a 7.5 MHz transducer. We have clarified this point in three ways. First, in the introduction (page 4) we’ve added the following clarification:

“The amplitude of this pressure wave is modulated at the multiheterodyne frequency f_i , which can be several orders of magnitude less than the comb repetition frequencies, f_{rep} . When the sample absorbs light from many pairs of comb teeth, at optical frequencies $\nu_0, \nu_1, \nu_2, \dots, \nu_n$, it produces a PA pressure wave with frequency components centered at the corresponding modulation frequencies $f_0, f_1, f_2, \dots, f_n$.”

Second, we moved the following discussion of the multiheterodyne frequency generation from the Methods section to the introduction (page 4):

“One advantage of DCPAS is that the bandwidth and center frequency of the generated acoustic signal, corresponding to the amplitude modulation on the optical signal, can be adjusted independently of the optical bandwidth and center frequency. This adjustment is made in a controlled fashion by tuning the frequency locking of the two combs to adjust $\nu_{2,0}$, $\nu_{1,0}$, and Δf_{rep} , and therefore, the multiheterodyne frequencies f_i . This allows any optical spectrum to generate a PA signal that can be detected by an ultrasound transducer despite the transducer’s limited bandwidth. In this demonstration, we use lasers with $\approx 160 \text{ MHz}$ repetition rates and set the difference in repetition rates (tooth spacings), $\Delta f_{rep} = f_{rep,2} - f_{rep,1}$, to be 66.81 Hz . This allows us to map the roughly 15 THz optical bandwidth (full width at -10 dB) into an acoustic bandwidth of $(15 \text{ THz}) \times (\Delta f_{rep} / f_{rep}) \approx 6.3 \text{ MHz}$ [32] while centering this modulation signal at the peak of the transducer responsivity, 7.5 MHz . In order to maintain this mapping as well as the optical coherence required for long term averaging of the signals, both combs are fully stabilized in the manner described in ref. [44].”

Third, we added Figure S3 with a caption that mentions that the multiheterodyne modulation frequencies can be chosen to match the transducer bandwidth and center frequency. The Figure S4 caption says:

“Figure S3. Transducer responsivity (blue dotted line) compared to the multiheterodyne frequencies generated by the dual-comb optical excitation (black solid line). Note that the agreement is not fortuitous. Rather, the relative repetition and offset frequencies of the two frequency combs are selected so that the multiheterodyne spectrum is matched to the transducer response. A transducer with a different bandwidth or center frequency could be accommodated by adjusting the frequency comb parameters. For instance, if the transducer spectrum narrows, one can similarly narrow the multiheterodyne spectral bandwidth by reducing the difference in repetition frequencies, assuming the two combs are sufficiently phase coherent.”

Reviewer 2: “The first time μ_a is mentioned in the formula of the photoacoustic pressure wave, it is stated as an absorption coefficient, it should however be absorption.”

We have more carefully defined the term μ_A as the absorption coefficient. We use this symbol consistently throughout the text. Below we list several changes we’ve made to ensure that μ_A is used consistently:

First mention in text (page 4):

“When the sample absorbs light from a pair of comb teeth at optical frequency ν_i , it produces a PA pressure wave with an amplitude, p , proportional to the amount of light absorbed by the sample (iii in Figure 1) [41–43]. This proportionality is given by $p \propto I_0(\nu_i) \times [1 - \exp(-\mu_A(\nu_i) \times L)] \times \beta$, where $I_0(\nu_i)$ is the intensity of the dual-comb excitation at optical frequency ν_i , $\mu_a(\nu_i)$ is the sample’s base-e absorption coefficient at optical frequency ν_i , L is the sample thickness that contributes to the PA signal, and β is a proportionality constant that depends on sample material and geometry.”

Equation 2:

“The measured PA response is proportional to the pressure wave’s amplitude multiplied by the transfer function of the transducer [47,48]. For weakly absorbing samples like the paraffin and PDMS in these experiments, the exponential in Beer’s law can be linearized, and we can write the measured PA response as

$$(2) \quad V_{PA}(\nu_{optical}) \approx \beta \times \mu_A(\nu_{optical}) \times I_0(\nu_{optical}) \times H_{transducer}(\nu_{optical})$$

where V_{PA} is the measured PA response as a function of optical frequency, $\nu_{optical}$, β is a proportionality factor to convert from sample absorption to PA pressure which may be material and geometry dependent, $\mu_A(\nu_{optical})$ is the sample’s optical absorption coefficient at optical frequency $\nu_{optical}$, $I_0(\nu_{optical})$ is the intensity of the optical excitation at optical frequency $\nu_{optical}$, and $H_{transducer}(\nu_{optical})$ is the ultrasound transducer’s frequency-dependent responsivity after scaling to optical frequencies according to equation (1).”

Equation 3:

“Thus, the normalized PA response of a sample is

$$\widehat{V}_{PA}(\nu_{optical}) = \frac{V_{PA}(\nu_{optical})}{I_0(\nu_{optical}) \times H_{transducer}(\nu_{optical})} = \beta \times \mu_A(\nu_{optical}), \quad (3)$$

Figure 4C and Figure 4D show the result of normalizing the polymers' PA responses according to equation (3). For comparison, these figures also show the optical absorbance spectra, $\propto \mu_A(\nu_{optical})$, of PDMS and paraffin measured in a transmission-mode spectrophotometer."

We've also re-labeled the right vertical axes in Figures 4C and 4D as "absorbance" instead of "absorption". References in the text to these figures have also been changed to "absorbance".

Reviewer 2: *"For a spectroscopist it is not correct to mention energies in wavelength and bandwidth in nanometers. The wavelength of light depends on the material in which it propagates, the light frequency not. So I suggest change/add this."*

We have removed references to wavelength in the manuscript and replaced them with frequencies.

Reviewer 2: *"The abbreviation PDMS is not explained the first time it appears in the manuscript."*

We have defined the abbreviation PDMS at its first use in the text on page 2.

Reviewer 2: *"Fig. 4 demonstrates the linearity of the optical power with the time dependent SNR of the photoacoustic signal. The latter is unclearly defined: a time dependent acoustic transient, such as Fig 2C? How is the SNR calculated from this?"*

The time-domain SNR is the peak height of the interferogram divided by the standard deviation of the signal far from the peak. This is defined in the Methods section (page 13). We have also added a discussion of the time-domain SNR calculation in Section 1 of the Supplementary Information.

From the Methods section (page 13):

"For the SNR calculations used in Figure 5, the time-domain SNR is defined as the peak height of the interferogram divided by the standard deviation of the noise outside of the interferogram centerburst (see section 1 Figure S2 in supplementary information)."

See Section 1 of the SI (including Fig. S2) for more information on the time-domain SNR calculation.:

"This same peak time-domain SNR is used in the main text in Figure 5 to investigate the scaling of the DCPAS SNR with power and integration time. In cases with very low SNR, for example for PDMS or paraffin at low comb powers or short integration times, the SNR can approach unity. In that case, the simple analysis above can lead to a bias because of the significant contribution of noise at the peak of the interferogram. To avoid this bias, we apply a matched filter based on the interferogram from a high-SNR measurement of the same sample to extract the peak interferogram voltage."

Reviewer 2: *"From Fig. 4 it is clearly seen that there is no saturation ongoing for the nanotubes, is this the same for the other materials used?"*

The absorbances are significantly smaller for both the PDMS and paraffin than for the VACNTs, thus we do not expect any saturation effects. However, because the PDMS and paraffin yielded small PA signals, it would be prohibitively difficult and time consuming to verify there is no saturation using the method we used for the VACNTs (decreasing the illumination intensity and checking that the PA magnitude decreased proportionately).

Reviewer 2: *"It is unclear why Figure S1 is supplementary information, this figure is needed to understand what I aimed for in the text."*

We have moved the experimental setup figure from the supplementary information to the main text. It is now Fig. 2 of the manuscript.

Response to Reviewer 3

Reviewer 3: *“In this manuscript, the authors demonstrate the first dual-comb photoacoustic measurement. Photoacoustic detection is widely used in sensitive spectroscopy of both gas-phase and condensed-phase samples. Photoacoustic spectroscopy has proved useful in spectral imaging systems, especially with samples that transmit poorly the optical wavelengths of interest. Dual-comb spectroscopy is an emerging technique to disperse the spectrum of a frequency comb, without a need for a separate spectrometer. This allows fast spectrum acquisition without mechanical limitations from a spectrometer, but often imposes heavy requirements to the stability of the light source itself. Additionally, dual-comb spectroscopy generally requires high-bandwidth detectors, if broad spectra are desired, which can be a challenge with acoustic detection. In this manuscript, the authors use a relatively high-bandwidth ultrasound detector, together with highly stabilized frequency combs, which can efficiently take advantage of the limited bandwidth of an acoustic detector.*”

My only concern with the manuscript is that the demonstrated measurement system has a rather low signal-to-noise ratio with reasonable averaging times, and it capitalizes poorly on the potential benefits of using dual-comb spectroscopy in comparison to, for example, a more traditional FTS measurement with a single comb. This is mainly due to using combs with low repetition rate in comparison to the required resolution. The high bandwidth acoustic detector also generally comes at a cost of sensitivity. The disadvantages have been partially counteracted by apodizing the interferograms, but in doing so, a large portion of the measurement time is effectively wasted. However, the authors clearly state that the reported measurements are a proof of concept, and the manuscript lays out practical steps required to improve the measurement and to properly reach the advantages offered by dual-comb spectroscopy. This mainly comes down to using combs with more appropriate repetition rate and wavelength. Such combs are less widely available and still come with plenty of challenges, but this demonstration of dual-comb photoacoustic spectroscopy would further motivate their development and optimization.

This demonstration that dual-comb technique can be applied with photoacoustic spectroscopy would be of high interests to a relatively wide reader base, including researchers in the fields of dual-comb spectroscopy, photoacoustic spectroscopy or spectral imaging, as well as researchers working to produce novel frequency combs, such as those with high repetition rates. The measurements and the results are described clearly and in high technical detail. In terms of the presentation I thought I should mention that when the manuscript was printed out, it appears that some parts of the figure caption texts appear with clearly varying shades of blue and the light grey trace used in figure 3A and B is practically invisible. All in all, the results are novel and convincing, and of high enough impact that I would support the publication of the manuscript.”

We thank the reviewer for their comments and agree with the reviewer that the SNR in this demonstration was low due to the low repetition rates of the combs. As the reviewer mentions, high power and high repetition rate combs would lead to significant improvements. We've expanded the discussion of these improvements as well as a range of other options for increasing the system SNR. Several changes have been added to the manuscript to clarify these issues. These changes are discussed in the responses to reviewers 1&2 above but are not repeated here. For instance, see responses 1 and 3 to reviewer #1 about how DCPAS SNR compares to conventional PAS and how higher repetition rate frequency combs could

improve SNR. Another option would be to use a more agile frequency comb with a variable repetition frequency, but this approach also requires improvements to frequency combs themselves. We thank the reviewer for pointing out the problems with the figure captions and have fixed these problems with the figure quality.

Additional changes

While considering the review comments, we noticed that the data plotted in Fig. 2 of the original manuscript were taken under slightly different illumination conditions than the other data in the original manuscript. To correct this, we replaced the data in Figure 2 of the original document (Figure 3 of the present revision) with data that were taken under the same illumination conditions as all of the other data in the manuscript.

We have also made minor wording changes at the very end of the second-to-last paragraph in an effort to clarify the issues related to normalization and the optical depth of the sample.

Finally, due to the addition of the OPO-PAS measurements, we have added an additional author.

REVIEWERS' COMMENTS:

Reviewer #1 (Remarks to the Author):

The authors have done good work when preparing the revised version of the manuscript. In particular, the additional experimental comparisons included in the paper nicely confirm the high potential impact of the work. I look forward to seeing the paper published in Nature Communications.

Reviewer #2 (Remarks to the Author):

After my first review of this manuscript, the authors replied satisfactory to my remarks and to the remarks of the other reviewers.
As such, I do not have further comments and I accept the manuscript for publication in the journal.